# Efficacy of the Treatment of Developmental Language Disorder: A Systematic Review

**DOI:** 10.3390/brainsci11030407

**Published:** 2021-03-23

**Authors:** Sara Rinaldi, Maria Cristina Caselli, Valentina Cofelice, Simonetta D’Amico, Anna Giulia De Cagno, Giuseppina Della Corte, Maria Valeria Di Martino, Brigida Di Costanzo, Maria Chiara Levorato, Roberta Penge, Tiziana Rossetto, Alessandra Sansavini, Simona Vecchi, Pierluigi Zoccolotti

**Affiliations:** 1Developmental Neurorehabilitation Service, UOC Infancy, Adolescence, Family Counseling, AULSS 6 Euganea—Padua Bacchiglione District, Via Dei Colli 4/6, 35143 Padua, Italy; 2Federazione Logopedisti Italiani, Via Daniello Bartoli, 00152 Rome, Italy; annagiulia.decagno@gmail.com (A.G.D.C.); tizzi.ross@gmail.com (T.R.); 3Institute of Cognitive Sciences and Technologies, CNR, 00185 Rome, Italy; cristina.caselli@istc.cnr.it; 4“Iuvenia” Rehabilitation Centre, C.da Piana, 86026 Morcone, Italy; valentina.cofelice@hotmail.it; 5Department of Biotechnological and Applied Clinical Sciences, University of L’Aquila, P.le S. Tommasi, 1, 67100 Coppito, Italy; simonetta.damico@univaq.it; 6CLASTA—Communication & Language Acquisition Studies in Typical & Atypical Population, Piazza Epiro 12D, 00183 Rome, Italy; chiara.levorato@unipd.it (M.C.L.); alessandra.sansavini@unibo.it (A.S.); 7Centro Panda, Via Antonio Gramsci, 38, 80022 Arzano, Italy; pinadc.1991@gmail.com; 8Health Professions Integrated Service, Azienda Ospedaliera dei Colli di Napoli, 80131 Napoli, Italy; valeria.dimartino@ospedalideicolli.it; 9Division of Phoniatrics and Audiology, Department of Mental and Physical Health, and Preventive Medicine, Luigi Vanvitelli University, Largo Madonna delle Grazie 1, 80138 Naples, Italy; dicostanzo.b@gmail.com; 10Cinetic Center, Neuromotor Rehabilitation Centre, Via Santella 26, 81025 Marcianise, Italy; 11Department of Developmental Psychology and Socialization, University of Padua, Via Venezia, 8, 35131 Padova, Italy; 12Department of Human Neuroscience, Sapienza University of Rome, Via dei Sabelli 108, 00185 Rome, Italy; roberta.penge@uniroma1.it; 13Department of Psychology “Renzo Canestrari”, University of Bologna, Viale Berti Pichat 5, 40127 Bologna, Italy; 14Department of Epidemiology, Lazio Regional Health Service, Via Cristoforo Colombo, 112, 00154 Rome, Italy; s.vecchi@deplazio.it; 15Department of Psychology, Sapienza University of Rome, Via Dei Marsi 78, 00185 Rome, Italy; pierluigi.zoccolotti@uniroma1.it; 16Neuropsychology Unit, IRCCS Fondazione Santa Lucia, Via Ardeatina 306, 00179 Rome, Italy

**Keywords:** developmental language disorder, intervention, evidence-based

## Abstract

Background. Language disorder is the most frequent developmental disorder in childhood and it has a significant negative impact on children’s development. The goal of the present review was to systematically analyze the effectiveness of interventions in children with developmental language disorder (DLD) from an evidence-based perspective. Methods. We considered systematic reviews, meta-analyses of randomized controlled trials (RCTs), control group cohort studies on any type of intervention aimed at improving children’s skills in the phono-articulatory, phonological, semantic-lexical, and morpho-syntactic fields in preschool and primary school children (up to eight years of age) that were diagnosed with DLD. We identified 27 full-length studies, 26 RCT and one review. Results. Early intensive intervention in three- and four-year-old children has a positive effect on phonological expressive and receptive skills and acquisitions are maintained in the medium term. Less evidence is available on the treatment of expressive vocabulary (and no evidence on receptive vocabulary). Intervention on morphological and syntactic skills has effective results on expressive (but not receptive) skills; however, a number of inconsistent results have also been reported. Only one study reports a positive effect of treatment on inferential narrative skills. Limited evidence is also available on the treatment of meta-phonological skills. More studies investigated the effectiveness of interventions on general language skills, which now appears as a promising area of investigation, even though results are not all consistent. Conclusions. The effectiveness of interventions over expressive and receptive phonological skills, morpho-syntactic skills, as well as inferential skills in narrative context underscores the importance that these trainings be implemented in children with DLD.

## 1. Introduction: Developmental Language Disorder

Language disorder is the most frequent developmental disorder in childhood [1]; however, it does not constitute a diagnostic category that refers to a homogeneous condition [2,3]. In some cases, the disorder is limited to production; however, in the most serious cases, it extends to the understanding of language. It can also affect different aspects of language processing, such as: (a) the form of language (phonetic, phonological, morphological, morpho-syntactic, and syntactic processing); (b) its content (semantic-lexical and phrasal processing); and (c) its use (pragmatic and discursive processing) [4,5,6]. Approximately 11–18% of children aged between 18 and 36 months [7,8,9] present a delay in the appearance of expressive language that, in the most severe cases, can also be observed in the receptive domain [10,11,12] in the absence of deafness, intellectual disability, brain injury, and cognitive disorder.

These children have been called late talkers [13,14]. The prognosis is generally good, as, in 70% of cases, expressive language improves significantly by three years of age and subsequently the development of language skills is generally in line with the expected performance in typical development [9,13,15,16,17,18]. However, some mild difficulties in daily communication interactions may persist [19,20]. Recovering children have been referred to as “late bloomers”. Thus, being a late bloomer does not necessarily imply a negative evolution; evidence indicates that the outcome is likely to be more favorable if the ability to understand language is preserved and there is no history of language and reading problems in the family [21,22]. However, even though many late speakers reach the same level of linguistic development as their peers, in 5–7% of the population the disorder persists after the age of three and a spontaneous recovery of language skills before school age is unlikely. In these cases, we speak of developmental language disorder (DLD) [4,6].

DLD has been defined as a neurodevelopmental disorder that includes a set of variegated clinical pictures that are characterized by delay or disorder in one or more areas of language development in the absence of cognitive, sensory, motor, affective, and important socio-environmental deficiencies [3,23]. The term DLD (e.g., Ref. [24]) or, more simply, Language Disorder, is now more commonly used instead of the more traditional “Specific Language Impairment” (SLI) [25,26,27], because it has been questioned whether language disorder is truly “specific” [28,29]. Indeed, it is well-known that a language disorder is frequently associated with various types of cognitive difficulties, which manifest in different ways, such as, for example, in procedural memory management [30], motor control [31], phonological working memory [32], and executive functioning [33]. Recently, the CATALISE Consensus [34] has chosen to use the term “developmental language disorder”, implying that it emerges in the course of development, rather than being acquired or associated with known biomedical causes. Although the term DLD is now frequently used [34], the terms ‘Primary Language Impairment’ and ‘Primary Language Disorder’ have also been used to account for the a-specificity of this language disorder and its unknown origin [24,35,36] (for further details, see the Method section). Thus, following the more recent international consensus [34], we will refer to language problems throughout the present systematic review in terms of DLD, regardless of how authors of previous papers, as reported in this review, named it.

The language difficulties of children with DLD often have severe consequences in pre-school and early primary school. In approximately 40 to 50% of cases, linguistic impairments lead to negative neuropsychological sequelae [9], particularly at the time of the change in expressivity from oral to written language, i.e., in the first two years of primary school, when literacy rests on the mapping of the phonetic system [37,38]. It has been shown that language disorder is associated with a high risk of school learning problems [39,40] (estimated as five times higher than in the general population [41,42]), behavioral and psychiatric problems [43,44], and disturbances in emotional and social adaptation [45,46]. Additionally, there is evidence that these problems persist in adulthood and throughout a person’s life, also affecting job opportunities [47,48,49].

The breadth of these problems and their negative impact on a child’s development indicates the importance of an early identification of children who risk of exhibiting DLD or related problems, with the potential benefit of promoting interventions in an age group in which significant improvement is most likely to occur [50]. Effective and early diagnosis can also facilitate the planning of targeted rehabilitation interventions before problems interfere with the formal education process [6,51].

### Interventions for the Developmental Language Disorder

Therefore, identifying effective interventions is a fundamental aim in clinical practice with children who have DLD. In fact, language intervention during development may not only have short-term outcomes on the language component treated, but also medium- and long-term influences on the global development path. The links between oral language acquisition and written language learning are well-known; also important are the negative consequences on the quality of social integration and the emotional development of children with language disorder (e.g., Ref. [44]). At the same time, research in this area is made complex by the need to identify modalities of intervention that, on the one hand, reflect the variability of the disorder in its components and in different age groups and, on the other hand, take the variables that intervene in determining stable and lasting changes in children with DLD into account.

The international literature offers a wide range of rehabilitation interventions aimed at children with DLD. They not only reflect the wide variability in the expression of language disorder at different ages, but indicate the importance of establishing which rehabilitation intervention provides the best care for children with DLD. The goal of the present study was to systematically analyze the effectiveness of interventions on children with DLD from an evidence-based perspective. This effort is not new. In particular, Law et al. [52] carried out a systematic review of the RCT studies on the effectiveness of language intervention. They found clearer results in the case of expressive phonological and vocabulary difficulties than in the case of receptive difficulties. Evidence on expressive syntax interventions was mixed, which indicated the need for further research. In fact, considerable research has been carried out since this review. Other reviews have focused on more particular issues, such as specific areas of language intervention (i.e., narrative-based interventions [53,54] and phonological and associated expressive language difficulties [55]), the mediating role of short-term memory over the efficacy of language intervention [56], or the use of videos and digital media in interventions that were carried out by parents [57]. As a considerable amount on new studies have been made available since the last systematic general review [52], it also seemed to be important to carry out an updated review of the literature. We also felt that this was timely since the described shift in perspective leading to an interpretation of language difficulties in terms of DLD (e.g., Ref. [24]).

Note that we focused on interventions on different aspects of language (e.g., phonetics/phonology; vocabulary; and, morphology/syntax) and domains (comprehension and production) and did not analyze studies that were concerned with the improvement of pragmatic skills. Most of the interventions deal with language production, but a few investigate interventions aimed at reception/comprehension. Some of the approaches in the literature adopted techniques and protocols that aimed at single components (minimal phonological pairs or particular morphological and syntactic deficit characteristics), while other approaches aimed at a wider and “ecological” stimulation of different aspects of language production.

The methods of administration depend, in part, on the approach, and can involve the intervention of specifically trained figures (speech and language therapists), educators, and teachers, or, increasingly, interventions mediated by parents with different training and supervision by clinicians. Another important aspect in evaluating the effectiveness of the interventions is the way the outcomes are evaluated. In fact, the tools that are used for diagnosis are often not very sensitive to change and the tools built ad hoc often only measure the skill being trained and do not allow for evaluating generalization to neighboring skills or other language domains.

The location of the intervention also varies according to the different approaches. In addition to the interventions that were carried out in the clinical setting, many interventions are carried out at school or at home (for a review, see [58]).

The frequency and duration of the interventions also appear to be very variable. Generally, individual interventions are tested in short and relatively low-intensity cycles, on very specific targets, and with the frequent absence of follow-up evaluations. Group interventions are more rarely described (both groups of children with DLD and the child with DLD within a typical developmental peer group).

Another element of complexity for the “evidence-based” identification of the effectiveness of treatments is the variability that is linked to the characteristics of the language. Therefore, it is necessary to carefully consider the applicability of the treatments (often developed in an Anglo-Saxon context) to other languages and the possible influences of the characteristics of our language on the effectiveness of the treatment itself.

Below, we report a systematic review (SR) of the studies on the effectiveness of intervention on children with DLD (including previous SRs, RCTs, and cohort studies).

## 2. Methods

A review of the literature was carried out as part of a Consensus Conference about the diagnosis and treatment of children with language disorder, which was held in Italy in November 2018 [36]. This Consensus Conference agreed to use the term ‘Primary Language Disorder’ for the Italian context, with it being clearer to clinicians in defining its origin not acquired or associated with a known biomedical cause, as detailed in the first paragraph of the Introduction. For this reason, the meaning of the term ‘Primary Language Disorder’, which was used in this Consensus Conference, corresponds to that of the term DLD, as recently adopted [34] and used throughout the present systematic review. The organization and implementation of this Consensus Conference followed the steps that were indicated in the Methodological Manual of the Italian Superior Institute of Health [59]. The review was prepared according to the Preferred Reporting Items for Systematic Reviews and Meta-Analyses (PRISMA) statement [60], and it includes a PRISMA flow diagram.

### 2.1. Selection Criteria

The clinical question was formulated while using the PICO approach and the criteria for inclusion and exclusion of studies were established a priori (see Table 1).

### 2.2. Source of Data and Screening

A systematic search of the literature published up until December 2020 was conducted through research on the following databases PubMed, Embase, Web of Science, The Cochrane Central Register of Controlled Trials (CENTRAL; 2020 Issue 11), SpeechBITE (speechbite.com; accessed on 30 December 2020), and PsycINFO (Ovid). In addition, we searched the clinical trials registers: ClinicalTrials.gov (clinicaltrials.gov accessed on 30 December 2020), World Health Organization International, and Clinical Trials Registry Platform (WHO ICTRP; who.int/trialsearch; accessed on 30 December 2020) for ongoing or unpublished trials on December 2020.

For each database a search strategy was developed by considering MESH terms and free terms (see Appendix A). In addition, further articles were identified by screening the reference lists of relevant reviews. Finally, experts and practitioners in the field, participants in the scientific technical committee, or working groups of the Consensus Conference indicated further potentially relevant studies.

### 2.3. Data Selection, Extraction and Quality Assessment

Titles, abstracts, and full text screening were performed by two independent reviewers. Disagreements after a full text review were resolved through discussion. Three independent reviewers extracted data from each included study. Information was extracted concerning: study design, population characteristics, type of test or treatment, type of comparison group, results, type of setting and figures involved, and the results of the studies. We did not contact the authors of relevant studies reporting incomplete data to request the missing information.

An evaluation of the quality and usability of the results of the reviewed studies was carried out by three independent reviewers.

The checklist “AMSTAR 2” [61] was used to evaluate the internal validity of systematic reviews. AMSTAR 2 is composed of 11 items: (1) “a priori” design; (2) duplicate study selection and data extraction; (3) comprehensive literature search; (4) the status of publication as an inclusion criterion; (5) list of studies; (6) characteristics of the included studies; (7) assessment of the scientific quality of the included studies; (8) use of the scientific quality in formulating the conclusions; (9) methods used to combine the findings of studies; (10) likelihood of publication bias; and, 11) conflict of interest).

For each criterion, the ‘yes’ (clearly done), ‘no’ (clearly not done), or ‘not clear or not applicable’ category was assigned. The revisions were then classified, as follows:from 8 to 12 criteria with the a “yes” assessment: high quality;from 4 to 7 criteria with the “yes” assessment: medium quality; and,3 or less criteria with the “yes” assessment: low quality.

For RCT studies, based on the criteria that were developed by the Cochrane Collaboration [62], the following dimensions were assessed:-random sequence generation (selection bias), which considers the risk that the allocation of subjects in the experimental and control groups may have occurred in a non-random way, indicating a possible problem in the selection of groups;-allocation concealment (selection bias), which evaluates the degree of protection against the risk that the trial operators are aware of the mechanism of random allocation of subjects;-the blinding of participants and personnel (performance bias), which considers the risk that the lack of blindness of trial objectives in participants and staff might alter performance (e.g., favoring a lack of expectations for the control group), thus affecting the trial outcome;-blinding of outcome assessment (detection bias), which indicates the risk that persons evaluating the study’s outcome are aware of the group assignment to different forms of intervention, thus influencing the probability of capturing the effects of the intervention;-incomplete outcome data (attrition bias), which evaluates the possibility that the presence of missing data modifies the estimation of the effects of interventions. The reasons for attrition or exclusion were reported as well as whether missing data were balanced across groups or related to outcomes; and,-selective reporting (reporting bias), which indicates the possible risk that only a selection of variables is presented in the report, e.g., the tendency not presenting measures for which the results were not significant (e.g., not presenting measures for insignificant results).

For each of the studies reviewed, an assessment of the possible presence of these bias risks is carried out. Figure 1 and Figure 2 summarize the results of the evaluation.

It is important to take the validity of the knowledge into account in order to assess the generalizability of the results of the included studies, i.e., the possible presence of a bias in the data, as well as transferability to clinical practice (external validity).

External validity was evaluated based on the transferability of results to clinical practice. For each of the RCT studies examined, the setting in which the study was carried out as well as the language covered by the intervention were assessed and are reported in Appendix B.

### 2.4. Data Synthesis

#### Analysis

The characteristics of the included studies are presented in tables and summarized narratively. A meta-analysis of outcomes was not appropriate due to the heterogeneity of the data; however, narrative results are presented.

## 3. Results

Using the bibliographic research, we identified 3334 reports after removing duplicates; two independent reviewers excluded 3219 reports on the basis of title and abstract. Any doubtful cases were resolved by discussion with a second reviewer.

We acquired 118 potentially relevant studies in full text, and we assessed their compliance with the a priori defined inclusion criteria. We identified 27 relevant studies [52,63,64,65,66,67,68,69,70,71,72,73,74,75,76,77,78,79,80,81,82,83,84,85,86,87,88] and excluded 91, which we considered to be ineligible. Figure 3 shows the PRISMA flow diagram for the selection of the studies.

### 3.1. Characteristics of Studies

Appendix B presents detailed tables with the characteristics of the included studies. Of the included studies, one was an SR [52] and 26 RCTs [63,64,65,66,67,68,69,70,71,72,73,74,75,76,77,78,79,80,81,82,83,84,85,86,87,88] (including two studies [86,87] derived from Loo et al.’s [89] review) on the effectiveness of interventions for the treatment of identified DLDs.

Law et al. [52] (score AMSTAR = 8) included 36 RCTs that evaluated different types of interventions that aimed at improving one or more of the following areas of language: expressive or receptive phonology, expressive or receptive vocabulary, and expressive or receptive syntax. In particular, the following comparisons were evaluated:interventions compared to no treatment or later treatments;specific interventions with respect to general stimulation conditions (e.g., studies in which children in the control group were assigned to conditions that were designed to simulate interaction in therapy without promoting the language area of interest. These are cognitive therapy, general play sessions or speech therapy that did not focus on the area of the specific linguistic deficit considered); and,interventions compared to other language therapy approaches (e.g., studies comparing what they considered a “traditional treatment” with what they considered to be an experimental treatment. The latter could be a different approach performed by the same person, such as “targeting early” against “late developing sounds”, or the same approach performed by different people, as in the case of “focused stimulation” provided by clinicians against that implemented by parents).

Of the 26 RCTs, six studies evaluated either expressive [63,64,65,66,67] or receptive [68] phonological skills, one on expressive vocabulary [69], eight grammar/morphological skills [70,71,72,73,74,75,76,77,78], with one reporting data on both phonological and morpho-syntactic skills [78], two on narrative skills [79,80]; two examined meta-phonological skills [81,82] and six language skills in general [83,84,85,86,87,88], including two [87,88] comparing the effectiveness of the “Fast ForWord Language” (FFW-L) training program with other intervention programs.

#### 3.1.1. Risk of Bias of Included Studies

With regard to the internal validity of RCTs, Figure 2 summarizes the data for the entire sample of 26 studies. Overall, the body of evidence was affected by a definitely high risk of bias in selection bias, since most of the studies did not report the method of randomization. Moreover, we judged most studies at unclear risk of bias of detection bias due to unclear blindness of participants and staff (24 out of 26 studies) and selectivity in the publication of results (19/26 studies).

#### 3.1.2. External Validity

Extrapolated data regarding the language targeted by the intervention and the context of treatment can be seen in Appendix B. Almost all of the included RCTs (21 out of 26) were conducted on the samples of English-speaking children and, in 16 studies, treatment took place in the school setting.

### 3.2. Effect of Intervention

Below are the studies identified according to the language area (outcome) being treated.

#### 3.2.1. Expressive Phonological Skills

Information regarding the effectiveness of interventions on phonological expressive skills in children with DLD comes from studies that were reviewed in the revision of Law et al. [52]. In particular, four studies specifically evaluated the effectiveness of several interventions on phonological expressive competence in children with DLD and reported an improvement of outcomes in the target children with DLD with respect to the control groups with no treatment (*N* = 264; SMD = 0.44, 95% CI: 0.01, 0.86) [50]. The effect was larger when treatments given by parents were excluded (*N* = 214; SMD = 0.67, 95%CI: 0.19, 1.16). Furthermore, the estimate was larger when only treatment lasting at least eight weeks was considered (*N* = 213; SMD = 0.74, 95%CI: 0.14, 1.33).

Five studies investigated the effectiveness of interventions on phonological expressive skills in children with DLD. A brief description of these studies follows (additional characteristics are reported in Appendix B).

Allen’s study [63] examined the effectiveness of two interventions, based on the use of maximally contrasting phoneme pairs, performed once or three times a week in children diagnosed with a speech sound disorder (SSD). Children were randomly assigned to one of three groups: one-time-per-week phonological intervention, three-times-per-week phonological intervention, and active control intervention, which was given a single-weekly treatment based on book narration. Based on the score for the percentage of correct consonants (PCC), children in the three-times per week group outperformed the single-weekly intervention group (and the control group) after eight weeks as well as after 24 weeks of training (when the overall training dosage was comparable). At a six-week follow-up, both of the experimental groups showed continued improvement without significant differences between them. Notably, the study did not consider language skills other than phonological ones.

In Lousada et al.’s [64] study, the effectiveness of a phonological therapy based on the combination of expressive phonological tasks, phonological awareness, and auditory discrimination and listening activities was compared to an articulation therapy that consisted of a traditional approach according to the “Van Riper Method”. At the end of the intervention, both groups showed improvements in verbal production, but children that were assigned to phonological therapy showed a greater improvement in the PCC score and a greater generalization of untreated words as compared to the other group.

In the study conducted by Diaz-Williams [65], the effectiveness of an intervention called “Gross Motor Activity” was examined: it is characterized by the production of target phonemes in single words, as depicted in four images in association with a motor activity (e.g., jumping). The group that was assigned to this intervention was compared with two other groups that received the “Structured Table Activities” and “Structured Table with Letter-Tracing Activities” interventions, respectively. In the two trainings, the same images were presented; however, in the first one, the words were simply incorporated in the table activities, whereas, in the second one, the children also received a card with the target sound in order to trace the target sound with their finger. All of the children showed a reduction in the average number of phonological errors on the HAPP-3 test [90] with no significant differences among trainings; moreover, in all cases, the intervention had a positive effect on the children’s homework.

In the study conducted by Wren and Roulstone [66], the effectiveness of a computer-supported therapy on phonological skills was evaluated. A group of children was presented with experimental software that mirrored board activities using interactive games and was compared with two groups: one group underwent desk therapy, which included a variety of games with images and objects and the other group received no treatment. The results showed no significant differences between the two groups in phonological production (measured with GFTA Sounds in Words subtest [91] and in terms of PCC), which was also confirmed in a follow-up three weeks after the end of the experiment.

Jesus et al., 2019 [67] evaluated the effectiveness of a 12-week novel tablet-based approach to phonological intervention targeting children with phonologically based speech sound disorders. A group of children (*N* = 22) was assigned to a combination of phonological awareness activities, phonological awareness program, auditory bombardment, and discrimination and listening tasks delivered with a tabletop or with an app running on a tablet. The results showed that both tabletop and tablet-based methods of delivery of a phonological intervention were effective in improving the speech of children. There was a significant improvement in PCC and in the percentage of phonemes correct from baseline (T1) to intervention (T3) for both groups, which was greater during the intervention period (between T2 and T3). Similar results were obtained for the percentage of correct vowel scores, with an improvement being noted at both baseline and intervention, but the increase after intervention was only significantly greater in the tablet group.

Overall, there is evidence that interventions aimed at expressive phonological skills produce appreciable results, even if it is not possible to specify which type of intervention is the most successful.

#### 3.2.2. Receptive Phonological Skills

Law et al. [52] showed no evidence of the effectiveness of interventions on phonological receptive skills. Only one study was identified that did not present significant differences between the groups.

We found one study that examined the effect of treatment over receptive phonological skills. Roden et al. [68] investigated whether the Auditory Stimulation Training with Musical material (ASTM) influenced auditory working memory, language processing, phoneme discrimination, and high frequency hearing abilities in preschool children with DPL (with low percentile ranges in the TROG-D [92]). Children in the experimental group heard acoustically modified music over earphones in small groups of 5–6 children. They revealed significant increases of working memory capacity measures, phonemic discrimination and speech perception at high frequency, and they outperformed control groups (pedagogical activities group, and no intervention groups) in all measurers.

Overall, the evidence on the effect of intervention on receptive phonological skills is too limited to draw any conclusion. Yet, the positive effects that were reported by Roden et al. [68] suggest the importance that further research will examine this linguistic area.

#### 3.2.3. Expressive Vocabulary

Law et al.’s review [52] reported the effectiveness of interventions that aimed at improving expressive vocabulary in children with expressive difficulties only when compared with no-intervention (*N* = 82; SMD = 1.08, 95% CI: 0.61, 1.55), but not when compared with other cognitive therapies (*N* = 25; SMD = 0.62, 95% CI: −0.24, 1.49). A large intervention effect was also present when parental reports of vocabulary were used as dependent measures (*N* = 136; SMD = 0.89, 95% CI: 0.21, 1.56).

A cross-over RCT study [69] evaluated the effectiveness of using e-books as a tool to support vocabulary acquisition in two experiments. The first experiment assessed whether the group of Dutch children with DLD (*N* = 29) was able to learn new words through reading electronic storybooks without the support of adults and whether storybooks with video and audio effects were more or less advantageous when compared to electronic versions with static illustrations (i.e., without effects); two stories that children had not heard during the intervention acted as a non-treatment control. The second experiment (*N* = 23) had a dual purpose, i.e., to confirm the results of the previous experiment and extend knowledge regarding learning new words in children with DLD by exploring two potential variables, i.e., phonological working memory and language skills. In the first experiment, better performance was obtained with “static” stories; this finding was also confirmed by the second experiment. Children with more severe DLD obtained less of an advantage from e-books when music and sounds were present (probably because they had difficulty in perceiving speech in noisy conditions).

Overall, there is still limited evidence that targeted interventions on expressive vocabulary acquisition produce effective results. It seems important that these partial results be confirmed in future RCT investigations.

#### 3.2.4. Receptive Vocabulary

No studies were identified that investigated the effectiveness of receptive vocabulary interventions in children with DLD.

#### 3.2.5. Morphological and Syntactic Expressive Skills

Law et al.’s [52] review reported seven studies that examined the effectiveness of different interventions on morphological and syntactic expression skills in children with DLD. Interventions proved to be effective when compared to non-interventions or other cognitive therapies, but the effect was only clear when children with severe language comprehension difficulties (*N* = 233; SMD = 1.02, 95%CI: 0.04, 2.01) were excluded.

Eight studies investigated the effectiveness of interventions on morphological and syntactic expressive skills in children with DLD. A brief description of these studies follows (also, see Appendix B).

The study by Plante et al. [70] aimed to evaluate the effectiveness of a treatment that used conversational recast for the correction of morphological and grammatical errors that were specific to the English language (such as past -ed, auxiliary -is, third person -s, possessive -s). A group of nine monolingual American children individually received the experimental intervention in the high variability condition (consisting of listening to the morpheme being treated in 24 verbs during each treatment session); a second group of nine children served as a control group and received the intervention in the low variability condition (based on listening to the target morpheme in 12 verbs, each restructured twice in each session). Children in the high variability treatment condition had better results and showed significantly better treatment effects for the target morphemes than the control group. However, the high variability condition produced a significant change in the use of trained, but not untrained, morphemes.

The study by Fey et al. [71] assessed the effectiveness of an intervention based on the Competing Sources of Input (CSI) hypothesis, which states that, at a certain stage of development, children cannot grasp the difference between subject-verb structures (SV) that appear in isolation and SVs that are part of a larger phrasal construction. In children with DLD, a delay in acquiring this grammatical rule is expected. The intervention, in English, was related to the development of verb morphology and concordance. Three treatment sections were provided for each target morpheme (the auxiliary “is” and the suffix of the third person singular/3S); past tense “-ed” was only monitored as a control. A control group of 11 children carried out a standard stimulation intervention (in which the comprehension activities were focused on semantic contrasts and the production activities included both declarative and interrogative stimuli) of equal duration and frequency. Both of the treatments were carried out, through individual sessions, by a researcher. The children assigned to the CSI group acquired greater skills in the use of “is”-“is” and in the understanding of “is-no”, as compared to the control group. Moreover, for the CSI group, a significant correlation between the understanding of “is-no” and the production of the auxiliary was observed. The difference between the two groups occurred, although the exposure to “is” was the same during the sessions; the authors found a strong support for the CSI. On the other hand, no significant differences emerged between the groups, either in the production of 3S or in the control -ed.

The study conducted by Smith-Lock et al. [72] evaluated the effectiveness of a grammatical intervention in which children in the experimental group received a cueing strategy, which, after an error, provides a hierarchy of facilitations that aimed at obtaining the correct response. The control group received an intervention characterized by a recasting reformulation strategy, in which, at the same time as the error, the correct target was given to the child without stimulating him/her to produce it. Both of the groups showed improvements in a series of tests [93] that examined grammar skills (use of pronouns he/she, past -ed, and possessives), but the effect was more evident in the “cueing” group. In the individual analysis, 50% of the children in the cueing group and 12% in the recasting group showed a significant effect of treatment. Finally, in an eight-week follow-up, there was no significant difference between the groups: in each group, half of the children who showed a significant gain in treatment retained it after eight weeks.

In the study by Washington et al. [73,74], the effectiveness of a computer program to improve morpho-syntactic expressive abilities (syntactic order of elements and morphological elements, such as the article “the”, the use of “-ing”, and the auxiliary “is”), was evaluated. In a first study [66], the experimental group was submitted to a computer-assisted (C-AT) program, called “My Sentence Builder”, which contained images aimed at facilitating the production of sentences and it was compared with two other groups: one group (nC-AT) was given desk activities with predetermined materials and a control group (NT) was given no treatment. The results showed that both of the interventions resulted in improvements in both the morpho-syntactic expressive skills (as assessed by the SPELT-P [94] tests) and spontaneous use of language when compared with no treatment (as assessed by the DSS [95] system). No significant differences were observed between the C-AT group and the nC-AT group. At the three-month follow-up, the treated groups showed better performance in grammatical competence as compared to the control group, while no difference was found between the two.

In a follow-up analysis of the same study, Washington et al. [74] examined the session-to-session progress in terms of efficiency (i.e., the first session in which the child achieved an 80% criterion) and syntactic growth (the individual advancement beyond basic sentence level) in the two groups of children who received the two forms of intervention. The Computer-Assisted Intervention group outperformed the Table-Top Intervention group for efficiency and syntactic growth.

The study by Yoder et al. [75] assessed whether the pre-treatment mean length of utterance (MLU) was able to predict which intervention, between the “Milieu Language Teaching” (MLT) and the “Broad Target Recasts” (BTR), was more valid in fostering the grammatical development of monolingual English children (aged between 30 and 60 months). MLT and BTR are both treatments that start from child-centered play and use recasts as a consequence of child utterances. MLT focuses on preselected grammatical targets, whereas BTR aims at any developmentally progressive grammatical structure on the basis of the actual utterances. Children who started the treatment with an MLU of ≤1.84 morphemes showed faster grammatical development if they underwent the MLT rather than the BTR treatment. No differences were found between groups in children with initial MLU >1.84 morphemes. Finally, most of the participants maintained grammatical growth after treatment. In fact, as a group, they showed a moderate gain in grammatical development between post-treatment assessment and a follow-up four months after the end of therapy.

Finestack and Fey’s study [76] compared the effectiveness of deductive and inductive techniques for learning new grammar skills. The deductive instruction, which was carried out with computer support, included a teaching session of the new grammatical morphology through modelling and an explicit auditory suggestion (“prompt”). In the following session, the researcher requested the production of the new grammatical morpheme with images and an explicit auditory prompt, followed by tests to evaluate the learning, generalization, and maintenance of the target morphemes. The inductive instruction provided the same intervention with the difference that the auditory prompt in the recast modelling and restructuring activities was implicit. The explicit approach to teaching new grammatical rules proved to be better than the implicit one. However, several limitations were found, as responses varied considerably between participants, rehearsal contexts, and sessions.

In a related study, Finestack [77] compared the effect of explicit instructions (aimed to make the learner aware of a given linguistic pattern) with a more traditional implicit approach in children with DLD. In particular, the acquisition, maintenance, and generalization of three novel grammatical forms (gender, aspect, and person targets) was examined after either a training with implicit instruction or a combined explicit-implicit (E-I) instructional approach. The results showed a greater proportion of pattern users (participants with a performance greater than or equal to 80% on a given probe) in the E-I group for the acquisition, maintenance, and generalization of the grammatical forms. The effect was clear when the data on grammatical forms were collapsed together and in the case of the gender target. These findings are in keeping with the idea of a greater effectiveness of interventions incorporating the use of explicit instructions to teach grammatical forms to children with DLD.

Only one RCT assessed the effectiveness of a combined morpho-syntactic and phonological intervention in children with both phonological and morpho-syntactic deficits. This was Tyler et al.’s study [78], in which two groups were compared to analyze the effects of such interventions on the non-target domain as well as possible variations in efficacy according to the sequence of interventions. The morpho-syntax group showed a significant improvement in both morpho-syntactic and phonological skills as compared to the control group, whereas the phonology group showed a significant improvement in phonological, but not morpho-syntactic skills when compared to the control group. No significant differences emerged in phonological and morpho-syntactic performance between the two treated groups. For both intervention sequences, greater changes were highlighted in phonological than in morpho-syntactic skills, but they were only of significant magnitude in the first group. Each type of intervention led to improvements in the treated domain, but the morpho-syntactic intervention also led to a change in phonological skills that are similar to that obtained by the first phonological intervention. Moreover, the sequence with morpho-syntactic, rather than phonological, treatment also resulted in slightly better overall morphosyntactic performance.

Overall, there is some evidence that interventions aimed at morphological and syntactic expressive skills in children with DLD produce effective results. However, a number of inconsistent results have also been reported, and it is not clear which factors drive these differential outcomes. Furthermore, all of the studies were conducted in English. Given the profound differences in morphological structure between English and other Indo-European languages (such as French, Spanish, and Italian, the main objective of the Consensus Conference from which the present review originates), it appears to be necessary that these results be supported by RCT studies conducted in a variety of languages before definitive conclusions can be drawn on this issue.

#### 3.2.6. Morphological and Syntactic Receptive Skills

In Law et al.’s [52] study, there are no indications of the effectiveness of interventions aimed at receptive syntax.

One of the studies already described [71] investigated the effectiveness of a training based on the “Competing Sources of Input” (CSI) hypothesis. It was also aimed at receptive grammar skills, in particular the understanding of questions with the present and past auxiliary (for a description of the study see the previous section). The results showed that understanding questions was better in children that were receiving a therapy based on the CSI hypothesis (with contrasts with respect to verb time) than in controls (where the stimuli are based on semantic contrasts). This hypothesis is closely linked to the acquisition of English grammar.

The information that is related to this area seems to be insufficient to draw any conclusions regarding the effectiveness of interventions aimed at improving receptive morphological and syntactic skills.

#### 3.2.7. Narrative Skills

Law et al.’s [52] review did not investigate the effectiveness of interventions on narrative skills.

Only two RCT studies specifically evaluated the effectiveness of a narrative skill intervention. One was Maggiolo et al.’s study [79], which assessed the effectiveness of a program aimed at stimulating narrative skills based on the formal organization and content of the narrative. The experimental intervention consisted of three phases, i.e., interaction activities with the child, development of the experimental program, and interactive storytelling. The experimental program was structured into five mini-programs, i.e., temporal relationships, causal and purpose relationships, story presentation, storytelling, and storytelling structure. In the experimental group, significant differences were observed before and after the intervention in both the content and form of the story. In particular, the performance in causal and temporal relationships during the organization of the narrative content significantly improved, while no pre-post intervention differences were observed for the purpose relationships. No significant differences were observed within the control group. Note that the two groups were not directly compared.

Dawes et al.’ study [80] aimed to develop, test, and evaluate a small-group intervention targeting oral inferential comprehension within a book sharing context for 5- to 6-year-old children with DLD. Inferential and literal comprehension were both measured using a new methodology of assessment. Children were randomly allocated to one of two intervention groups: inferential comprehension group (ICI: *N* = 19) or phonological awareness control group (PA: *N* = 18). The mean comprehension scores prior to intervention were not significantly different for the two groups of children. When compared to the control PA group, the participants in the ICI group demonstrated a significant increase in the inferential comprehension scores from pre- to post-intervention, which was maintained over time. In addition, the ICI group scored significantly higher than the PA group for inferential comprehension on a post-intervention generalization measure. The results also demonstrated significant improvements at the individual level. No significant difference between the two groups for literal comprehension scores emerged at any assessment point.

The available information is still limited, but both studies examined reported clear increases in the comprehension of causal and temporal links after the intervention. Therefore, it appears to be important that these findings be confirmed and substantiated in further research.

#### 3.2.8. Meta-Phonological Skills

The Law review [52] did not investigate the effectiveness of interventions on meta-phonological skills. Of the RCTs included, two specifically evaluated the effectiveness of different interventions on meta-phonological skills in children with DLD. A brief description of the individual studies follows.

The study conducted by Hesketh et al. [81] evaluated the effectiveness of specific training on meta-phonological skills through awareness tasks with phonemes and syllables. Specifically, the tasks first focused on syllables and rhymes, then on the recognition of the first and last phoneme of the word, and, finally, on the phonological manipulation of adding or deleting phonemes in the word. The children in the experimental phonological awareness (PA) intervention were compared to a control group that received a language stimulation (LS) program with activities of linguistic comprehension, knowledge of writing, verbalization of emotions, and development of vocabulary and semantics. No significant difference was found between the two groups for rhyming knowledge; on the contrary, a difference emerged regarding the ability to isolate, segment, and manipulate phonemes, as well as to add and suppress phonemes, in favor of the PA group. However, the results should be interpreted with caution, because of the large variability within the experimental group (e.g., for the two most advanced tasks, segmentation and addition/suppression, only a small minority of children showed improvements). Furthermore, only children with an adequate cognitive level showed that they benefited from the intervention: in fact, cognitively weaker children did not benefit, even after an intensive period of intervention.

Hund-Reid and Schneider’s study [82] evaluated the effectiveness of training on phonological awareness and grapheme-phoneme correspondence in preschool children. The intervention that was chosen for this study was the “Road to the Code”. This is a phonological awareness program for young children [96]. It is based on principles that include, in each session, explicit teaching of one or two types of phoneme manipulations (e.g., initial sound isolation and/or initial sound identification) and fusion and segmentation, as well as sound-symbol awareness activities (manipulation of phonemes with letters). The experimental group showed significantly greater improvement than the control group on the measures of phonemic fluency, phonemic segmentation, and non-word fluency. These gains were maintained, even one month after the intervention. Other aspects were also evaluated, such as the knowledge of writing and speed of reading letters (that were not among the outcomes defined in the research protocol carried out for this Consensus Conference) for which no significant differences were found.

Overall, there is still limited information on the effectiveness of interventions on meta-phonological skills in children with DLD; however, the results of the two reviewed studies indicate this as a potentially interesting area of intervention. Further work is warranted, possibly also examining languages other than English.

#### 3.2.9. General Language Skills

Law’s review found only one study that aimed at training general linguistic skills with overall non-significant results.

Six of the RCTs included assessed the effectiveness of different interventions on language skills in general. A brief description of the individual studies follows.

In Roberts and Kaiser’ study [83], the effect of the “enhanced milieu teaching” (EMT) intervention that was carried out by parents (specifically trained by therapists and educators) on receptive and expressive language skills was evaluated. The intervention included four phases: setting the basics for communication; shaping and broadening communication; time delay strategies; and, finally, prompt strategies. The children in the treatment group showed higher gains in both expressive and receptive vocabulary than the control group. A comparison with the typically developing children showed that both groups with DLD continued to have significantly poorer language skills. However, when compared to the untreated children, the treated ones managed to grow at rates similar to those of children with typical development during the intervention. These results were considered to be preliminary, since the group size was not only small, but was a sub-sample of a larger study. Long-term outcome measures were also lacking.

Wake et al.’s studies [84,85] assessed whether an intervention on a population of four-year-old Australian children with language deficits could improve language outcomes and associated outcomes. Two-hundred four-year-old children already included in two previous studies (“Let’s Learn Language” and “Let’s Read” [97,98]) were selected (179 actually completed the study—91 in the experimental group and 88 in the control no-treatment condition). The children in the experimental group benefited from a home intervention aimed at promoting narrative skills, vocabulary, grammar, phonological awareness, and pre-reading skills, with a program that included 18 sessions distributed in three blocks of six weeks. This was characterized by: (a) a short review of the previous week; (b) activities introduced by the researcher directed towards the child; (c) activities for parents and children to be carried out together with the support of the researcher; and, (d) activities for home practice. The parents were then asked to talk to the child adopting specific language, to use a storybook, and to write down the activities in a diary. The control group, on the other hand, did not carry out any intervention.

At the five-year evaluation [84], significant improvement was found in the experimental group when compared to the control group for phonological awareness and graphemic recognition, but not for verbal production and understanding. A very positive perception by parents and a favorable cost–benefit ratio emerged. At a six-year assessment [85], improved language skills were found in both groups without significant inter-group differences. A significant improvement in phonological processing skills remained in the experimental group. In Wake et al.’s study [85], the authors reported that it was possible to implement relatively low-cost interventions with non-specialized personnel. However, no evidence emerged that this intervention actually improved outcome more than typical development. Finally, limitations also emerged: only a small number of families were at a disadvantaged socio-economic level and most of the language drops were mild. Therefore, it is not clear what results would have emerged in the case of children with a more compromised linguistic background.

The Wilcox et al.’s study [86] tested the efficacy of the Teaching Early Literacy and Language (TELL) curriculum [99] that was provided by preschool teachers. Ninety-one teachers were randomly assigned to TELL curriculum or Business-as-usual (BAU) contrast condition. Children with speech and language impairment in the experimental classes received, with the whole class, supportive and explicit teaching practices for oral language and early literacy skills. The authors did not find significant differences in the performance between TELL and BAU classes at a standardized assessment of receptive and expressive skills, phonological processing and awareness, and letter knowledge, but only in the curriculum-based measures (oral language and early literacy skills targeted in TELL program). They observed that BAU teachers also provided vocabulary and early literacy instruction and they concluded for the TELL efficacy for improving targeted oral language and early literacy skills.

Two studies [87,88] were derived from Loo et al.’s [79] review. The purpose of Gillam et al.’s study [87] was to determine whether a computer instructional program that was designed to improve auditory temporal processing skills (Fast ForWord-Language—FFW-L [100]) was more effective than other types of intervention for improving language and auditory processing in children with DLD. Two-hundred and sixteen children were randomly assigned to one of the four arms of intervention (see Appendix B for details). The children in all four arms made significant improvement in auditory processing, receptive and expressive language measures from pre- to post-testing, as well as in the follow-up three and six months later. Nevertheless, the children who received FFW-L did not do better than children in other interventions of equal intensity and primary outcome.

In Cohen et al.’s study [88], seventy-seven children with receptive-expressive specific language impairment were randomly assigned to Fast ForWord intervention [100], an alternative computer-based intervention or to a control group. After six weeks of computer games exposure at home, the improvements in expressive and receptive language performances of children in the experimental groups did not exceed those of the control groups.

Some of the studies on the effectiveness of different interventions on general language skills are now available, and they indicate this as a promising area of investigation. However, results appear to be mixed with several studies showing negative results.

## 4. Discussion

The main aim of the present review was to identify the most effective treatments to adopt for children with DLD. Because most of the studies aimed to verify the effectiveness of interventions for specific language skills, the analysis of the literature was organized according to the target language area. In this regard, the results of the new RCT studies that were identified after Law et al.’s [52] review partially confirm the indications already published and allow formulating some new hypotheses of effectiveness.

The interventions aimed at phonological, lexical, morphological and syntactic expressive skills are those most studied, presumably because these verbal components are easily identifiable and during treatment can be isolated from other aspects of language. The proposed activities are limited to these specific skills and the changes obtained can be verified through standardized scales of measurement. In addition, children with DLD who have a language profile that is limited to these skills are the most numerous. They are easily identified by non-specialist healthcare staff, school staff, and caregivers, and therefore represent a high percentage of access and demand for care by specialist services.

From the analysis of the studies included in this review, we can derive some general indications regarding the treatment of children with DLD. We have verified the effectiveness of intensive interventions based on the treatment with pairs of maximally contrasting phonemes, auditory discrimination, and phonological awareness. Regarding the phonological expressive component, a direct intervention, limited in time but intensive (i.e., three times a week), which includes auditory discrimination activities and it is based on contrast in traits, can bring significant improvements [63,64] that are maintained in the medium term.

Additionally, with regard to morpho-syntax, some of the strategies can be considered effective, such as recast or reformulation of the child’s production by the adult in conversation [70]. The same strategy is also proposed in tasks of storytelling that involve retelling stories [71]. Other valid strategies are that of cueing, i.e., providing suggestions to the child to try to stimulate the correct production [72], and that of the auditory prompt, i.e., explicit suggestions that are related to the grammatical rule [76,77]. Despite the effectiveness of these strategies, analysis of the studies has not allowed us to derive useful indications to define which of these could be the most appropriate in the area of treatment.

Furthermore, some of the studies have provided support for the effectiveness of treatments for meta-phonological and narrative skills [79,80,81,82]. The results appear to be related to the progress of research on the role of these skills in the language development of children with DLD.

New studies have also emerged that investigate interventions that aimed at developing general communicative language competence. These interventions are carried out at home, are mediated by parents, and are under the supervision of the clinician [83,84,85]. The interesting indication in these studies is that they are not only aimed at younger children with language delay, but that the indirect intervention also continues for children with DLD. These interventions are aimed at encouraging that more language skills be carried out in the contexts of the child’s life (at home and/or at school), so they can be considered to be more ecological, and the results could be easily generalized. The very recent study conducted by Wilcox (86) reported the efficacy of a teaching curriculum provided by teachers in improving oral language and early literacy skills of preschooler children with DLD. This study shows that even a systematized teaching program conducted in the school setting by teachers can promote a general improvement in communicative language abilities of preschool children with DLD, even if such an improvement is not evidenced at a standardized assessment of different areas of language. Still, one must add that results on general language skills are mixed, with several studies showing inconclusive findings (e.g., [87,88]). Thus, while the reviewed studies raise the interest in this type of intervention, the overall picture of findings is too scattered to be able to draw a firm conclusion on the effectiveness of intervention on general language skills.

Relatively little evidence of the strategies or techniques aimed at improving receptive language skills has been identified. One possible interpretation of the lack of results could be related to the complexity of interventions that aimed at verbal comprehension, i.e., it is difficult to isolate the individual receptive skills and free them from more general skills, such as pragmatic skills or semantics, creating activities that aimed at specific receptive skills in the rehabilitation setting. In addition, although children with a disorder that also affects language comprehension have a more severe clinical picture than children with only expressive DLD and a worse prognosis, they are numerically fewer. Therefore, it may be difficult to find enough children for an RCT study. Finally, the intervention may be less (or not) effective, also because, in some cases, weaknesses or deficits in comprehension (particularly lexical, morphosyntactic, and narrative) are associated with weak (even if within low-normal limits) cognitive skills.

Still, some recent studies have shown promising results. Regarding phonological skills, Roden et al. [68] demonstrated that the Auditory Stimulation Training with Musical material significantly improves working memory, phonemic discrimination, and speech perception in preschoolers with DPL. In evaluating the efficacy of an intervention targeting oral inferential comprehension within a book sharing context for preschoolers with DLD, Dawes et al.’ study [80] reported significant and sustained improvement in the group of children with DLD in inferential skills. Overall, the evidence on the effect of intervention on receptive phonological and on the comprehension of narratives skills is too limited to draw firm conclusions, but the quoted studies underscore the interest in further pursuing this area of research.

With respect to the other variables that are necessary to provide indications for treatment, such as the setting, frequency and duration of interventions, the way the results are evaluated, the age of the child, and the long-term effects of the intervention, the studies taken into consideration do not allow for us to draw firm conclusions. This is due to both the extreme heterogeneity across studies and the fact that these variables are often not described explicitly and exhaustively by the authors.

The risk of bias was high in some studies and unclear in most studies (particularly bias due to method of randomization, blinding of participants and personnel, and selectivity in the publication of results), thus decreasing the certainty in results. The fact that several studies examined relatively small samples of children is also of note; this may be problematic, particularly in the case in which a more comprehensive linguistic deficit is present (i.e., both expressive and receptive), which is typically associated with larger individual differences. These considerations indicate the importance and urgency that standards of RCT reporting will be improved in future research.

Regarding the transferability of the results to different languages as compared to the one in which the intervention is delivered, it appears that the specificity of the phonological repertoire and of the morphological and syntactical rules always deserves a reflection on the implicit differences between languages. Thus, for some language levels, it is possible to generalize the results, while, for others, it is necessary to modify the verbal stimuli of the treatment by adapting them to the linguistic context while maintaining techniques and strategies.

## 5. Implications for Clinical Practice and Research

The reviewed evidence highlights the importance of carrying out timely assessments of linguistic (and, in particular, phonological and morphosyntactic) skills in pre-school children, so as to provide, if necessary, targeted treatment before the start of primary school, while considering the importance of these skills for future school learning.

## 6. Limitations

There seems to be two general problems in analyzing the literature on interventions on DLD. On one hand, there are many more studies available for expressive than receptive disorder. We have noted above that this, in turn, may be due to the asymmetry in targeted children between these two types of disturbances. Yet, there is a risk here to consider that the interventions on expressive disorder are more effective, simply because more studies were carried out. Only further research on receptive disorder will allow for us to reach more definite conclusions on this point. On the other hand, most reviewed studies concern English and research on other languages is scattered. Accordingly, there is a need for systematic research in a wide range of other languages, particularly in the areas (such as morpho-syntax) where linguistic differences are more marked.

## 7. Conclusions

The present systematic review provides up-to-date information on the effectiveness of linguistic interventions for children with DLD. Evidence indicates that early intensive interventions in three to four-year old children are effective in the area of phonological expressive skills with acquisitions being maintained in the medium term. Some effectiveness of interventions over morpho-syntactic skills and, to some extent, on meta-phonological and narrative skills, was also detected. By contrast, there is fewer evidence that interventions on phonological receptive skills or receptive vocabulary are effective, but some recent studies raise the interest in further pursuing this area of research. A number of trainings aimed at general linguistic skills; the results are mixed, which makes it difficult to draw a definite conclusion, although there are indications that this may be a promising area of further investigation. Most research is carried out in English-speaking children, indicating the importance of studies in other languages.

Overall, information on linguistic interventions is quite different, depending on the linguistic skills investigated, indicating the need of further RCT studies in this area. Nevertheless, the currently available information indicates the importance of implementing timely assessments of linguistic skills and, whenever appropriate, targeted treatment.

## Figures and Tables

**Figure 1 brainsci-11-00407-f001:**
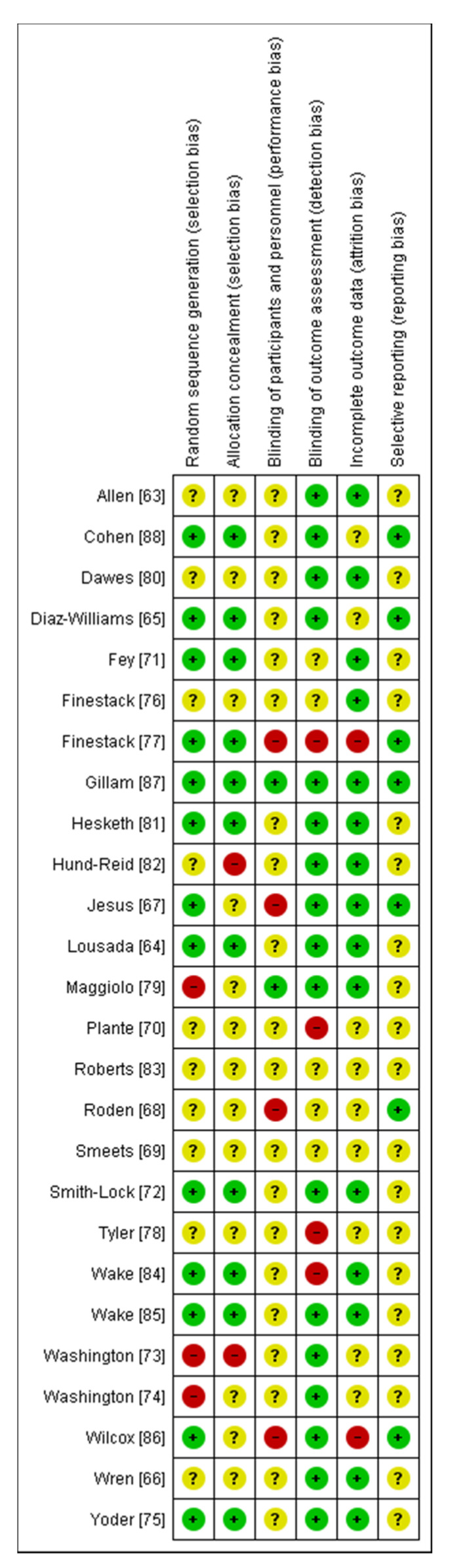
Risk of bias found in individual studies considered in the present review. Green: low bias risk; yellow: unclear bias risk; red: high bias risk [63,64,65,66,67,68,69,70,71,72,73,74,75,76,77,78,79,80,81,82,83,84,85,86,87,88].

**Figure 2 brainsci-11-00407-f002:**
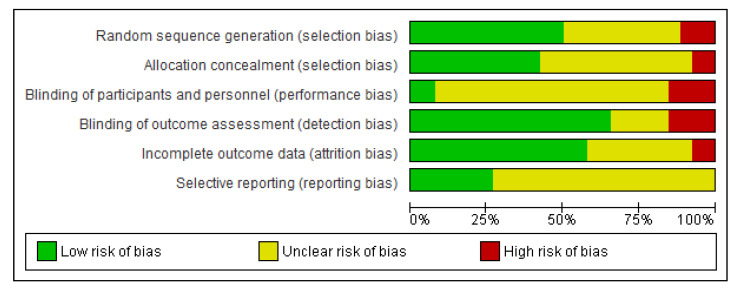
Percentages of risk of bias found in the studies considered in the present review.

**Figure 3 brainsci-11-00407-f003:**
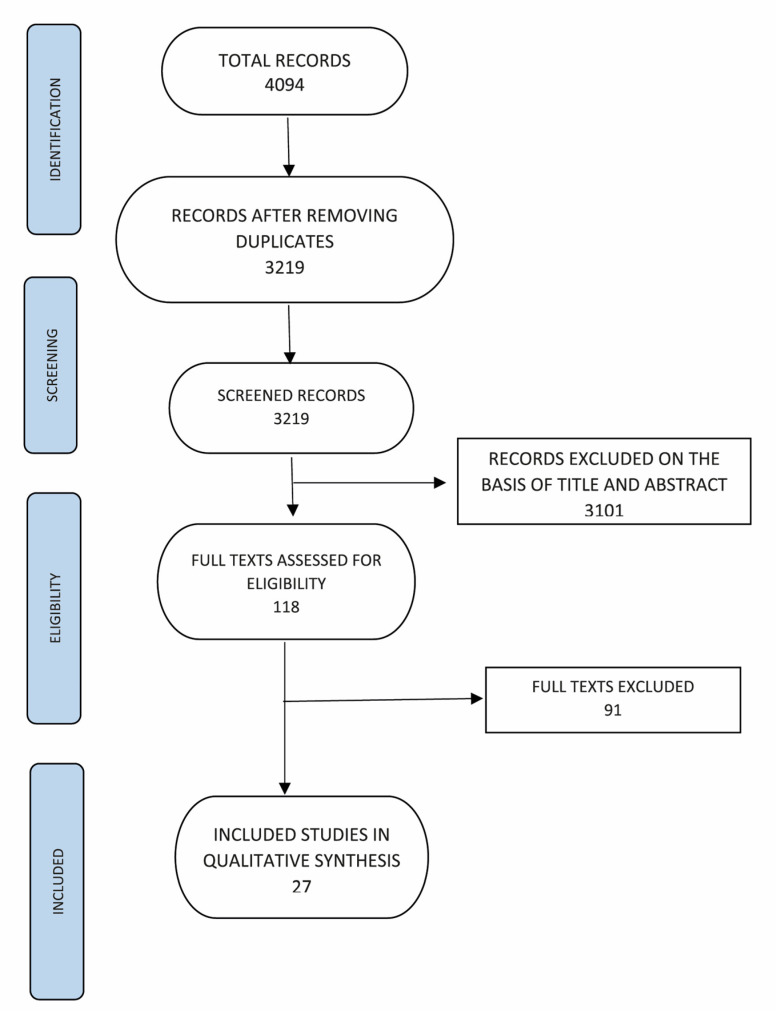
Prisma flow chart of included studies.

**Table 1 brainsci-11-00407-t001:** Inclusion and exclusion criteria.

	INCLUSION CRITERIA
POPULATION	Preschool and primary school children (up to 8 years of age) diagnosed with Developmental Language Disorder (DLD)
INTERVENTION	Any type of intervention that aims to improve the child’s skills in the phono-articulatory, phonological, semantic-lexical and morpho-syntactic fields. The intervention can be administered at the individual or group level, by different types of professional figures (teacher, health care personnel, parents, speech therapists, other health care professionals), with different durations and frequencies, in different settings (home, clinics, community, school).
COMPARISONS	Other types of experimental interventions, waiting list, no intervention, other interventions that are considered “usual care”.
OUTCOMES	Improvement in language expression and reception in the areas of semantics, syntax and phonology Social behavior Adverse events (such as parental anxiety) Dropout
SETTING	Any setting
STUDY DESIGN	Systematic reviews (SR) or meta-analysis of randomized controlled trials (RCTs), RCTs. If no RCTs are available: cohort studies. We considered only SR that (1) searched at least one database; (2) reported its selection criteria; (3) conducted quality or risk of bias assessment on included studies; and (4) provided a list and synthesis of included studies. SRs that identified observational studies were included if results from RCTs were provided separately.
LIMITS	No temporal or language limits
EXCLUSION CRITERIA	Children with cognitive delay, deafness, autism spectrum disorders, genetic syndromes (Down syndrome, Klinefelter syndrome), neurological deficits, pervasive developmental disorders, traumatic brain injuries, primary disorders (sensory, neurological, psychiatric), children with dysphonia, dysarthria, dysrhythmias or stuttering, dyslalias or specific speech articulation disorder, bilingualism. Commentaries, opinions, editorials and studies that do not report a quantitative synthesis of the association between intervention and outcome measures.

## Data Availability

Not applicable.

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
