# Peer review of "Efficacy of the Treatment of Developmental Language Disorder: A Systematic Review"

_brainsci, 2021, doi:10.3390/brainsci11030407_

Round 1

Reviewer 1 Report

This paper is a systematic review of the evidence for interventions for primary language disorder (or DLD). The topic and scope are definitely of interest to clinicians in the field, and this paper could provide a helpful resource to clinicians wanting to get an good overview of the current evidence base. The review includes evidence from a previous review, but adds 21 studies in addition to the previous SR, so I do believe it is adding value beyond previous reviews of the literature. The paper uses tools and standards for SRs, including PRISMA, Cochrane risk of bias measures, and the quality appraisal tools for SRs AMSTAR.

A lot of my comments pertain to errors in the manuscript – these should be easily rectified, but does suggest the paper would benefit from a thorough check before resubmission. I present my comments in rough order as to the part of the manuscript they relate to.

Introduction, second and third paragraphs: these do not fit together very well. The point seems to move back and forth here. Should these be one paragraph together?

Why not use the term Developmental Language Disorders? This is the term that has largely replaced “Specific Language Impairment”, rather than PLD? You do mention DLD, but I am interested in why you haven’t opted to use this term? The authors may already be aware of the work that has been published outlining why DLD is the preferred term in the field: Bishop, D. V., Snowling, M. J., Thompson, P. A., Greenhalgh, T., Catalise‐2 Consortium, Adams, C., ... & house, A. (2017). Phase 2 of CATALISE: A multinational and multidisciplinary Delphi consensus study of problems with language development: Terminology. Journal of Child Psychology and Psychiatry58(10), 1068-1080.

The remit of the review seems to focus on structural language interventions rather than pragmatic ones. This is not a problem, but it might be worth stating this in your introduction somewhere. Or did you look for pragmatic interventions and not find anyway?

Extra period in the heading 1.1. Interventions for primary language disorders

Line 131 – Not sure “Furthermore” is what you mean here?

In your list of factors for consideration, in lines 136 to line 143, “modality of outcome evaluation” appears twice.

Line 143 – I am not sure what you mean by language type? Do you mean Italian versus English versus German, for example.  When I got to lines 168-171 it is clear that is broadly what you mean -  perhaps refine what is said in the list so it is clearer.

Line 174 – “The introduction should briefly … details on references” – this is clearly text left in from a template document/instructions for authors and needs removing.

This review is including previous SRs on more specific topics, RCTs, and cohort studies. I think it’s worth laying this out in the main text before getting to the data extraction sections – I missed it and was rather confused when I got to the this section and had to revisit Table 1 a few times before I was sure I had understood.

Line 276 – I think RS should be SR

The heading “Characteristics of Studies” is repeated

The two additional Loo studies (mentioned in lines 300-301 and again in lines 326-327) – I think from what you’ve written it sounds like the original papers of these studies weren’t accessible but you’ve included what you can based on their description in the review? That’s fine, but make it explicit that the originals couldn’t be accessed, if that is the case (otherwise it’s not clear why they didn’t get included in analyses/Figs etc).

Line 684 – allow *for the* formulating *of* some…

Lines 725-728: “it is more difficult to make a differential diagnosis compared to other developmental disorders” – could you say a bit more here, and there published evidenced that this is the case, that children with severe receptive problems are harder to tell apart from say children with global intellectual impairment?

Lines 738-Lines 742 – I think you could argue here that reporting in the field needs quite a bit of improvement. The level of unclear reporting around key areas of bias seems really troublesome to me. You note the problems but why not claim here that we need better RCT reporting standards in the field of PLD?

Author Response

This paper is a systematic review of the evidence for interventions for primary language disorder (or DLD). The topic and scope are definitely of interest to clinicians in the field, and this paper could provide a helpful resource to clinicians wanting to get an good overview of the current evidence base. The review includes evidence from a previous review, but adds 21 studies in addition to the previous SR, so I do believe it is adding value beyond previous reviews of the literature. The paper uses tools and standards for SRs, including PRISMA, Cochrane risk of bias measures, and the quality appraisal tools for SRs AMSTAR. Response: we thank the reviewer for appreciation of our manuscript. A lot of my comments pertain to errors in the manuscript – these should be easily rectified, but does suggest the paper would benefit from a thorough check before resubmission. I present my comments in rough order as to the part of the manuscript they relate to. Response: below we detail the changes made with regard to the specific comments of the reviewer. Introduction, second and third paragraphs: these do not fit together very well. The point seems to move back and forth here. Should these be one paragraph together? Response: the first three paragraphs have been re-arranged.  We hope that the new presentation proves clearer. Why not use the term Developmental Language Disorders? This is the term that has largely replaced “Specific Language Impairment”, rather than PLD? You do mention DLD, but I am interested in why you haven’t opted to use this term? The authors may already be aware of the work that has been published outlining why DLD is the preferred term in the field: Bishop, D. V., Snowling, M. J., Thompson, P. A., Greenhalgh, T., Catalise-2 Consortium, Adams, C., ... & house, A. (2017). Phase 2 of CATALISE: A multinational and multidisciplinary Delphi consensus study of problems with language development: Terminology. Journal of Child Psychology and Psychiatry, 58(10), 1068-1080. Response: we generally agree with the comments of the reviewer. During the Consensus conference on which the present analysis is based, it was agreed to use the term “primary 
language disorder” instead of “specific language impairment”.  However, admittedly, this choice was in large part based on considerations specific of the Italian clinical setting.  While we maintained the expression PLD in our first version of the manuscript, we do agree with the reviewer that “Developmental Language Disorder” is nowadays more widely used. Accordingly, we have switched to this term in the revision of the paper. In particular, we state (lines ca. 98 and following: … Following the more recent international consensus [34], we will thus refer to language problems throughout the present systematic review in terms of DLD, regardless of how authors of previous papers, reported in this review, named it.” The remit of the review seems to focus on structural language interventions rather than pragmatic ones. This is not a problem, but it might be worth stating this in your introduction somewhere. Or did you look for pragmatic interventions and not find anyway? Response: You are correct.  Pragmatic interventions were out of the scope of our analysis.  This concept has been added to the introduction (ca. line 147 in the revised paper).  Extra period in the heading 1.1. Interventions for primary language disorders Response: the error has been fixed. Line 131 – Not sure “Furthermore” is what you mean here? Response: the sentence has been amended.  In your list of factors for consideration, in lines 136 to line 143, “modality of outcome evaluation” appears twice. Response: Sorry; the error has been corrected.   Line 143 – I am not sure what you mean by language type? Do you mean Italian versus English versus German, for example.  When I got to lines 168-171 it is clear that is broadly what you mean -  perhaps refine what is said in the list so it is clearer. Response: We actually meant “typology of languages”; however, in the revised version of the manuscript, we have simplified this part of the text and the sentence was deleted. Line 174 – “The introduction should briefly … details on references” – this is clearly text left in from a template document/instructions for authors and needs removing. Response: sorry; we apologize for this error. The part has been removed from the text. 
This review is including previous SRs on more specific topics, RCTs, and cohort studies. I think it’s worth laying this out in the main text before getting to the data extraction sections – I missed it and was rather confused when I got to the this section and had to revisit Table 1 a few times before I was sure I had understood. Response: This information has been added at the end of introduction. Line 276 – I think RS should be SR Response: Thank you; we have corrected the error The heading “Characteristics of Studies” is repeated Response: Sorry; we have deleted the repetition. The two additional Loo studies (mentioned in lines 300-301 and again in lines 326-327) – I think from what you’ve written it sounds like the original papers of these studies weren’t accessible but you’ve included what you can based on their description in the review? That’s fine, but make it explicit that the originals couldn’t be accessed, if that is the case (otherwise it’s not clear why they didn’t get included in analyses/Figs etc). Response: Yes.  However, we have been now able to examine these two texts and the description of the studies have been added in the paper (as well as in the figures).  Line 684 – allow *for the* formulating *of* some… Response: the correction has been made.   Lines 725-728: “it is more difficult to make a differential diagnosis compared to other developmental disorders” – could you say a bit more here, and there published evidenced that this is the case, that children with severe receptive problems are harder to tell apart from say children with global intellectual impairment? Response: Actually, in revising out text, we chose to omit this sentence which was indeed not very clear. Lines 738-Lines 742 – I think you could argue here that reporting in the field needs quite a bit of improvement. The level of unclear reporting around key areas of bias seems really troublesome to me. You note the problems but why not claim here that we need better RCT reporting standards in the field of PLD? 
Response: Thank you for this suggestion on which we certainly agree. A consideration along these lines has been added to the text.

Reviewer 2 Report

Introduction

The variables of interest are reasonably well defined, and the problem is stated clearly.  The authors need to do a better job of situating this review with respect to prior reviews.  The text related to this issue on lines 122 – 133 is less than satisfying.  The reader needs specific information about the content and conclusions of prior reviews along with a clear explanation of the need for the yet another review.

Lines 144 – 172 read like a list of aspects that the interventions vary on.  A careful explanation of the definition of “intervention” and the limits of that definition would be more helpful and more interesting.

Lines 174 – 182 read like they were pulled from a paper on how to write systematic reviews.

Method

It is now 2021, and the literature that was reviewed does not extend past 2017.  Therefore , the review is not as current as it should be.  The review needs to be updated to at least include papers published in 2019 and preferably 2020.

I don’t recall seeing a reliability assessment to demonstrate that multiple raters coded the articles in the same way. 

I don’t recall seeing information about the extent of missing data.

Results

It makes sense to categorize the students with respect to the outcomes that were measured. 

In the discussion, there are multiple comments about the effectiveness of intervention strategies such as recasts, cuing, auditory prompts.  A table of effect sizes for such intervention strategies would help the reader more readily see the important data that support the authors’ conclusions.

The authors concluded that it was important to provide targeted treatment to preschoolers.  But I didn’t see a report of the compelling evidence to support that.

Discussion

Line 708 – it would be helpful if the authors elaborated on the strategies that were effective.  I think we’re past the point of just saying that we can effectively intervene on certain skills.

Line 769 – The most current paper that was reviewed was from 2017.  Therefore, this review is not up-to-date.

Author Response

Introduction

  1. The variables of interest are reasonably well defined, and the problem is stated clearly. The authors need to do a better job of situating this review with respect to prior reviews. The text related to this issue on lines 122 – 133 is less than satisfying. The reader needs specific information about the content and conclusions of prior reviews along with a clear explanation of the need for the yet another review. Response: We have expanded this part of the text by describing more explicitly the study by Law et al. (2003). Furthermore, at stated below we have enlarged the literature search up to 2020 as requested.  Accordingly, we also found additional reviews that have been quoted.  Overall, as requested, we have expanded the description of previous reviews on related areas. We hope that the presentation proves clearer.
  2.  Lines 144 – 172 read like a list of aspects that the interventions vary on. A careful explanation of the definition of “intervention” and the limits of that definition would be more helpful and more interesting. Response: in carefully rereading our text we agree with the reviewer that the list of characteristics of trainings was not highly informative and we have deleted it from the revised version.  We also revised our general presentation of intervention distinguishing between techniques and protocols aimed at single components and approaches aimed at a wider and "ecological" stimulation.
  3. 3.  Lines 174 – 182 read like they were pulled from a paper on how to write systematic reviews. Response: Sorry; this part was not necessary and has been deleted in the present revision.
  4. 4. Method 
    It is now 2021, and the literature that was reviewed does not extend past 2017.  Therefore, the review is not as current as it should be.  The review needs to be updated to at least include papers published in 2019 and preferably 2020. Response: We understand the problem. Our original version was based on the outcome of a Consensus Conference run in Italy and the dates of the reviewed studies were associated to this.  As requested, we have updated our search using exactly the same criteria as before and we are now able to present an analysis updated to 2020.  Furthermore, we have updated our comments and considerations (particularly in the case of narrative skills and general language skills). We hope that the new version appears more updated and convincing.
  5. 5. I don’t recall seeing a reliability assessment to demonstrate that multiple raters coded the articles in the same way. Response: The selection and quality evaluation was separately carried out by three authors. This information was added to the revised text.
  6. 6. I don’t recall seeing information about the extent of missing data. Response: Actually, we did not contact authors of relevant studies reporting incomplete data to request for missing information.  This information was added to the text  
  7. 7. Results It makes sense to categorize the students with respect to the outcomes that were measured.  Response: sorry: we are not sure to understand exactly what the reviewer means with this sentence. 

    8. In the discussion, there are multiple comments about the effectiveness of intervention strategies such as recasts, cuing, auditory prompts. A table of effect sizes for such intervention strategies would help the reader more readily see the important data that support the authors’ conclusions. Response:  we carefully examined this proposal.  In examining the relevant studies, we have come to the conclusion that it would be relatively complex to make such a table.  Please note that an additional consideration is related to the quite short time which was given for the revision which was largely spent in updating the literature search 
    9. The authors concluded that it was important to provide targeted treatment to preschoolers. But I didn’t see a report of the compelling evidence to support that. Response: The sentence has been revised and simplified deleting the reference to pre-school intervention.
  8. 10. Discussion Line 708 – it would be helpful if the authors elaborated on the strategies that were effective. I think we’re past the point of just saying that we can effectively intervene on certain skills. Response: We have re-worded the sentence indicated and hope that the version is clearer.  However, we would also like to add that, at the present time, it is still difficult to exactly pinpoint the exact best approaches to be used in each area of intervention.
  9. 11. Line 769 – The most current paper that was reviewed was from 2017.  Therefore, this review is not up-to-date. Response: We have updated our search as requested by the reviewer and we feel that now this expression can indeed be maintained. 

Round 2

Reviewer 2 Report

The authors have done an acceptable job of responding to earlier comments and criticisms.  The addition of some current papers means that the review is now up-to-date, and it is reasonably informative.